# Effect of CuO Loading on the Photocatalytic Activity of SrTiO₃/MWCNTs Nanocomposites for Dye Degradation under Visible Light

Xuan Truong Mai [1], Duc Nguyen Bui [1],*, Van Khang Pham [1], Thi Ha Thanh Pham [1], Thi To Loan Nguyen [1], Hung Dung Chau [2] and Thi Kim Ngan Tran [2],*

1   Faculty of Chemistry, Thai Nguyen University of Education, No. 20 Luong Ngoc Quyen Street, Thai Nguyen City 24000, Vietnam
2   Institute of Applied Technology and Sustainable Development, Nguyen Tat Thanh University, Ho Chi Minh City 70000, Vietnam
*   Correspondence: ducnguyen@tnue.edu.vn (D.N.B.); nganttk@ntt.edu.vn (T.K.N.T.)

**Abstract:** In this study, we report on the preparation of copper oxide/strontium titanate/multi-walled carbon nanotube (CuO/STO/MWCNTs) nanocomposites and their photocatalytic activity for degradation of dye under visible light. The crystalline structures of the nanocomposites were investigated by an X-ray diffraction (XRD) technique, which explored the successful fabrication of CuO/STO/MWCNTs nanocomposites, and the cubic STO phase was formed in all samples. For the morphological study, the transmission electron microscope (TEM) technique was used, which had proved the successful preparation of CuO and STO nanoparticles. The energy dispersive X-ray spectroscopy (EDX), dark field scanning transmission electron microscope (DF-STEM-EDX mapping), and X-ray photoelectron spectra (XPS) analysis were performed to evidence the elemental composition of CuO/STO/MWCNTs nanocomposites. The optical characteristics were explored via UV–Vis diffuse reflectance spectroscopy (DRS) and photoluminescence (PL) techniques. These studies clearly indicate the effect of the presence of CuO and MWCNTs on the visible absorption of the CuO/STO/MWCNTs nanocomposites. The photocatalytic activity of CuO/STO/MWCNTs nanocomposites was evaluated by the degradation of methylene blue (MB) dye under visible light irradiation, following first-order kinetics. Among the different x% CuO/STO/MWCNTs nanocomposites, the 5 wt.% CuO/STO/MWCNTs nanocomposites showed the highest photocatalytic efficiency for the degradation of MB dye. Moreover, the 5% CuO/STO/MWCNTs showed good stability and recyclability after three consecutive photocatalytic cycles. These results verified that the optimized nanocomposites can be used for photocatalytic applications, especially for dye degradation under visible light.

**Keywords:** CuO/SrTiO₃/MWCNTs; visible light; photocatalysis; degradation

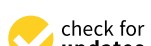

## 1. Introduction

Commercial dyes have been extensively employed in many industrial fields, such as textile, foodstuff, and leather, etc., and they have played a crucial role in industrial effluents. Nevertheless, the majority of these dyes are reported toxic, pose a potential carcinogenic risk to nature, and are considered a threat to the environment, due to their removal from industrial effluents. As a matter of fact, a variety of methods, such as adsorption, biodegradation, filtration, electrochemical, precipitation, coagulation, ion exchange, and solvent extraction, have been applied to eliminate and degrade the dye removals [1]. However, these approaches have several drawbacks, such as the formation of the production of toxic sludge or hazardous byproducts, slow destructive and high production of sludge, incomplete removal of ion, and the replacement of pollutants from one phase to another [2].

As a result, photocatalytic degradation using a semiconductor as the photocatalyst is attracting more attention from researchers as a solution to environmental problems [3]. The mechanism of photocatalytic involves the photogenerated electrons and holes. Upon light irradiation, electrons are excited and jump from the valence band (VB) to the conduction band (CB), leading to the creation of holes in the VB of catalysts. Then, the photogenerated electrons migrate to the surface of the catalyst to participate in redox reactions with other chemical species to form a variety of active species ($\cdot OH, \cdot O_2$, and $H_2O_2$). Finally, these active species can effectively degrade and mineralize the pollutants. In these photocatalytic processes, the high recombination of the photogenerated electron–hole pairs remarkably limits the photocatalytic efficiency of catalysts [4].

Regarded as the most effective photocatalysts, perovskite-type $SrTiO_3$ (STO) has been introduced as the most investigated compound for photocatalytic applications, including water splitting, $H_2$ evolution, inorganic/organic pollutant, and dye degradation [5]. Nevertheless, the STO band gap value of 3.2 eV (equivalent to λ of 380 nm) leads to the sole activation in UV light range, which corresponds to about 3–5% of the solar total spectrum [6]. Moreover, the pure STO photocatalytic ability has been reported to be quite low, owing to the high recombination rate of photogenerated electrons and holes. These challenges lead to a decrease in the photocatalytic efficiency and impede extensive application progress by solar light utilization of STO [7]. Hence, it is vital that the solar energy utilization be expanded and the rapid charge recombination be minimized through the STO modification. To strengthen the photocatalytic efficiency of STO in visible light utilization, techniques such as the doping of metal ions, anion doping, loading of metal, and coupling of two semiconductors with large and small band gap energies have been examined and become available [8]. Among them, coupling with other semiconductors is an attractive approach to achieve a more efficient separation of photogenerated electrons and holes, allowing for the strengthening of photocatalytic efficiency. In general, when both semiconductors (or only one of them) in the composite photocatalysts are simultaneously activated by light, the photogenerated electron would transfer from the semiconductor having a more negative CB to the semiconductor with a more positive CB. Meanwhile, the hole would transfer from the semiconductor with a more positive VB to the semiconductor with a more negative VB. Therefore, the efficient separation of photoexcited electrons and holes is achieved, which effectively improves the photocatalytic activity of the catalysts [9].

It is believed that CuO is a narrow band gap semiconductor (Eg = 1.7 eV). According to previous reports, the pure CuO exhibits negligible photocatalytic activity [10]. In addition, CuO has been employed to enhance the photocatalytic activity for some semiconductors with wide band gaps [11]. The CB and VB values of CuO are located at 0.55 eV and 2.07 eV, while those of STO are at −1.32 eV and 2.20 eV, respectively. Since the CB of STO is higher than that of CuO, the photogenerated electrons can easily transfer from STO to the surface of CuO [12]. The electrons in the VB of STO are excited into the CB under light irradiation, which leads to the creation of holes in the VB of STO. Then, the photogenerated electrons in the CB of STO will transfer to the surface of CuO, which improves the separation of the photogenerated electron–hole pairs. Therefore, the photocatalytic performance of STO is improved [9]. In fact, several researchers have synthesized visible photocatalyst systems based on STO and CuO. Choudhary et al. have synthesized CuO/STO bilayered films using a sol–gel method, which were used for a water splitting reaction. As was mentioned in the literature, the bilayered system supports enhanced photocatalytic efficiency, which is associated with the improvement of conductivity and separation of photogenerated electron–hole pairs at the CuO/STO [13]. Recently, Ahmadi et al. [14] presented the synthesis of CuO/STO nanoparticles by employing a combination of precipitation–deposition and impregnation methods. Then, the photocatalytic activity of the CuO/STO was investigated by the photodegradation of Rhodamine B dye under UV light irradiation. In our previous studies, Cu and CuO were also used as co-catalysts to enhance the photocatalytic efficiency of STO for $H_2$ evolution from methanol solution [15,16]. It was indicated from the results

that Cu and CuO could facilitate the separation of the photogenerated electrons and holes, which leads to an improvement of the photocatalytic efficiency of the STO nanoparticles.

It is widely accepted that carbon nanotubes (CNTs) have caught much attention in various areas, thanks to the uniqueness of their characteristics including: structure, chemical, thermal, electrical properties [17]. In fact, the semiconductor oxides, such as $TiO_2$ and ZnO, supported by the CNTs have been introduced for photocatalytic organic pollutant degradation [18–24]. Specifically, CNTs have been stated to enhance the photogenerated electron–hole lifetime, as the junction developed at the interface of CNTs-$TiO_2$ facilitates the separation of photogenerated electrons and holes [25]. Further, the combination of CNTs and $TiO_2$ creates extra band gap states in $TiO_2$, due to the formation of C–O–Ti bonds, leading to extend its absorption to visible light [26]. Besides, CNTs provide a large surface area to concentrate the adsorption of pollutants, thereby bringing them to the surface of $TiO_2$ to carry out the photocatalytic reaction [27]. Recently, to promote the photogenerated electron–hole pairs and minimize the recombination rates, graphene has been documented in STO. The photocatalytic activity of STO–graphene nanocomposites has been evaluated through the degradation of acid orange 7 (AO7) under UV irradiation. The results revealed that the photogenerated electrons were rapidly captured by graphene, thus leading to the enhancement of separation of electron–hole pairs for the photocatalytic reaction [28]. Similar to graphene, MWCNTs possess unique properties, including large surface areas, high aspect ratios, tubular hollow cavity, and mechanical, thermal, electrical, and nano-sized stability, leading to the enhancement of their usage in photocatalytic applications. Due to its outstanding properties, MWCNTs have been used as ideal support form nanocomposites with improved performances in photocatalytic applications. In fact, the combination of graphene or MWCNTs with other catalysts is proven to be an efficient way to improve the separation of electron–hole pairs, which leads to the strengthening of the photocatalytic efficiency of catalysts.

Recently, Suresh Sagadevan et al [29] synthesized CuO loaded on reduced graphene oxide (rGO) nanocomposites. According to the report, the CuO/rGO nanocomposites showed excellent photocatalytic activity for dye degradation. Based on the above discussion, it can be seen that the combination of CuO, STO nanoparticles, and CNTs may provide an ideal photocatalytic system for dye degradation. However, research on the CuO/STO/MWCNTs nanocomposites photocatalytic performance has rarely been conducted. On the other hand, methylene blue (MB) is usually used in textile industries. This dye comprises colored, nonbiodegradable compositions that are harmful to living organisms. Therefore, the removal of dyes from wastewaters is very important [3]. In this paper, we present the facile synthesis of a novel photocatalytic system containing CuO, STO, and MWCNTs. The photocatalytic activity of the photocatalytic system has been studied through the photodegradation of MB dye under visible light irradiation. The influence of CuO contents on the properties and photocatalytic activity of the nanocomposites has been investigated in detail.

## 2. Results and Discussion

### 2.1. Characterizations of As-Prepared Samples

The crystallographic structure and composition of the CuO/STO/MWCNTs were determined by XRD and depicted in Figure 1. For comparison, MWCNTs and pure STO were also included as the reference. It can be seen that XRD patterns consisted of two peaks at 25.8° and 43.1°, corresponding to the (002) and (100) planes of MWCNTs (Figure 1a), respectively [30,31]. The diffraction peaks at 32.4°, 39.9°, 46.4°, 57.8°, 67.8°, and 77.2° were indexed as (110), (111), (200), (211), (220), and (310) planes of cubic STO (JCPDS 35-0734), respectively (denoted with (❖) symbol). On the other hand, as shown in Figure 1b, the STO prepared by the sol–gel method was shown to contain a small amount of $SrCO_3$ impurity (denoted with (●) symbol). It is accepted that $SrCO_3$ is a common product in the preparation of STO. The diffraction peaks of $SrCO_3$ were located at around 25.2°, 25.9°, and 44.2° (JCPDS 05-0418) and occurred in its XRD diffractogram [32]. For the nanocomposites,

the main peaks of the CuO/STO/MWCNTs were in good accordance with pure STO. It can be seen that the intensity and the width of the diffraction peaks of $SrCO_3$ in the CuO/STO/MWCNTs (Figure 1c–e) were higher and wider than that of the $SrCO_3$ phase in STO (Figure 1b). This is explained by the fact that the diffraction peaks of the MWCNTs and the $SrCO_3$ are overlapped, due to their proximity, leading to the increase of the intensity and the width of the diffraction peak of $SrCO_3$ phase in STO. Therefore, the peaks were at $2\theta = 25.1°$, $25.8°$ in the XRD patterns of CuO/STO/MWCNTs, which were attributed to the characteristic peaks of the MWCNTs and the $SrCO_3$ phase in STO (denoted with (*) symbol). Additionally, the peaks at $2\theta = 36.1°$, $49.1°$, $54.1°$ were attributed to the characteristic peaks of the CuO (according to JCPDS card no 05-661, the diffraction peaks of CuO were placed at around $27.54°$, $29.56°$, $35.65°$, $38.79°$, $48.8°$, $53.71°$, $58.37°$, $61.71°$, and $66.08°$ (denoted with (❖) symbol) [33]).

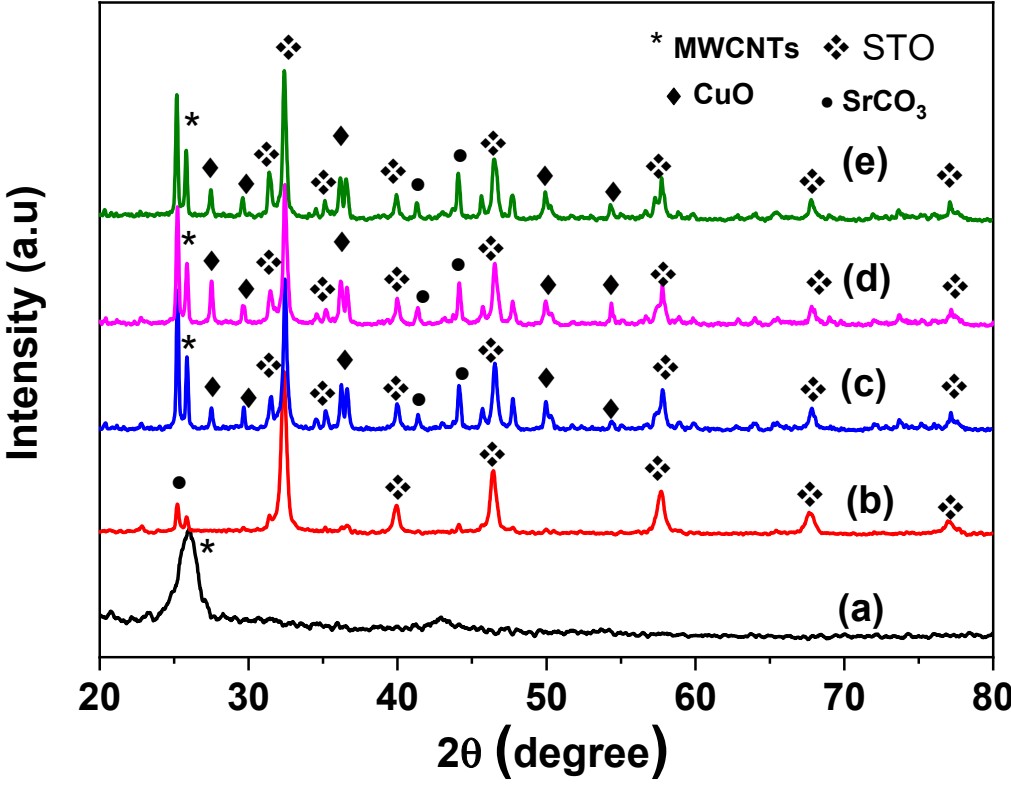

**Figure 1.** XRD patterns of MWCNTs (**a**), STO (**b**), 1% CuO/STO/MWCNTs (**c**), 3% CuO/STO/MWCNTs (**d**), and 5% CuO/STO/MWCNTs (**e**).

The elemental compositions of the samples were explored through EDX analysis. EDX plots of STO are shown in Figure 2a. One can observe only peaks related to Sr, Ti, and O in Figure 2a, thus expressing the elemental purity of pure STO. On the other hand, signals related to Cu, C appeared in the EDX plot of CuO/STO/MWCNTs, as shown in Figure 2b–d. The evidence for the composition of Cu in CuO/STO/MWCNTs nanocomposites has been proven in the EDX plots.

To further evidence the presence of CuO in the nanocomposites, the 5% CuO/STO/ MWCNTs sample was selected to perform XPS analysis. The survey spectra and high-resolution scans of the 5% CuO/STO/ MWCNTs samples are displayed in Figure 3. As shown in Figure 3a, the sample only included Cu, C, Sr, Ti, and O. This was consistent with EDX results. In Figure 3b, spin orbital splitting photoelectrons of Cu $2p_{3/2}$ and Cu $2p_{1/2}$ were placed at 933.6 and 952.6 eV, which corresponded to CuO [34,35]. From Figure 3c, the respective binding energies of Ti $2p_{1/2}$ and Ti $2p_{3/2}$ were located at 464.0 and 458.2 eV. The two respective binding energies were assigned to typical $Ti^{4+}$ [36]. The binding energies of Sr $d_{3/2}$ and Sr $d_{5/2}$ were 134.4 and 133.5 eV, respectively (Figure 3d). The two bands were

assigned to typical $Sr^{2+}$ [37]. The high-resolution spectra of C 1s shows that the C 1s peak centered at 284.6 and 288.7 eV, which corresponds to C–C $sp^2$ and O–C=O, respectively (Figure 3e) [38]. The results obtained from XPS analysis confirmed the element Cu in the nanocomposites is in the form of CuO.

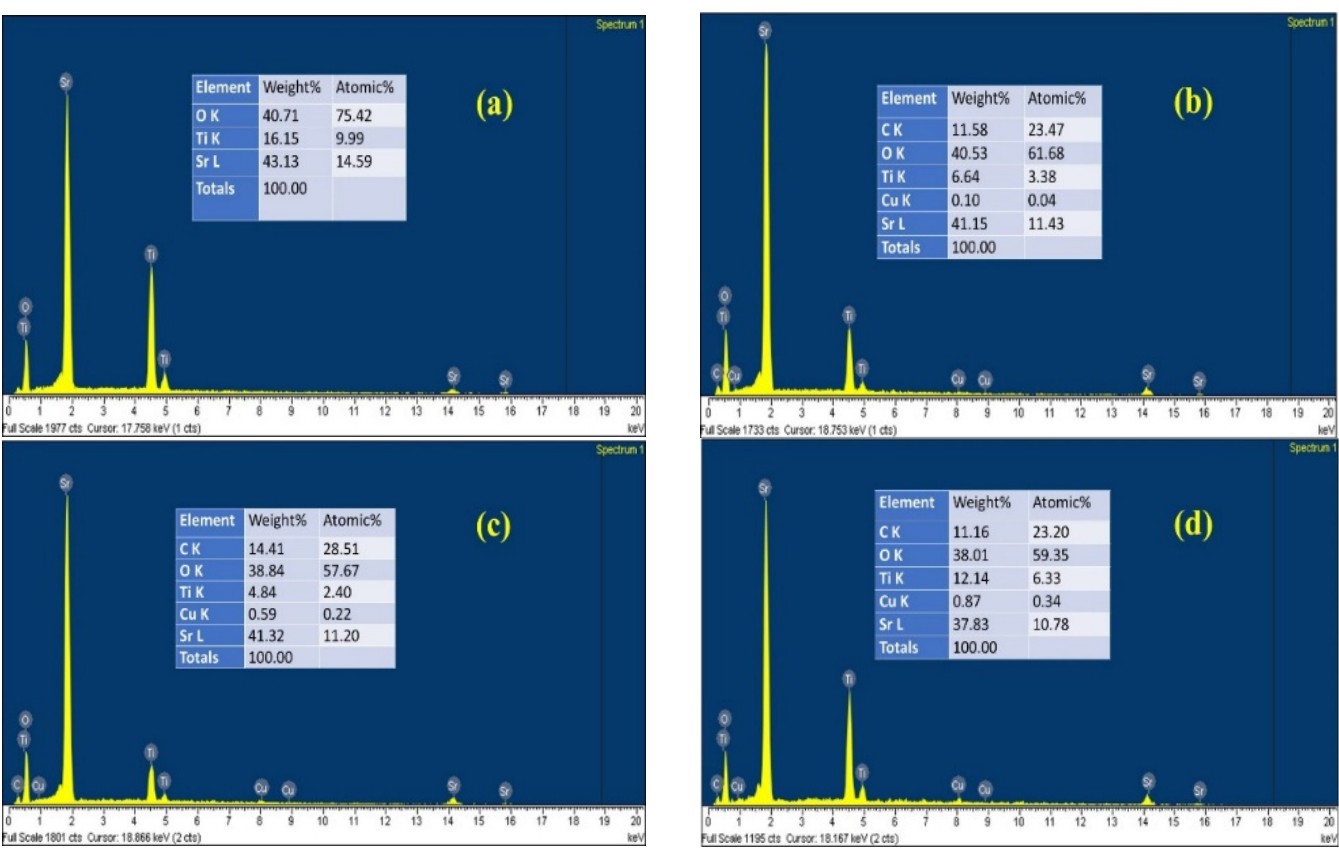

**Figure 2.** EDX spectra of STO (**a**), 1% CuO/STO/MWCNTs (**b**), 3% CuO/STO/MWCNTs (**c**), and 5% CuO/STO/MWCNTs (**d**).

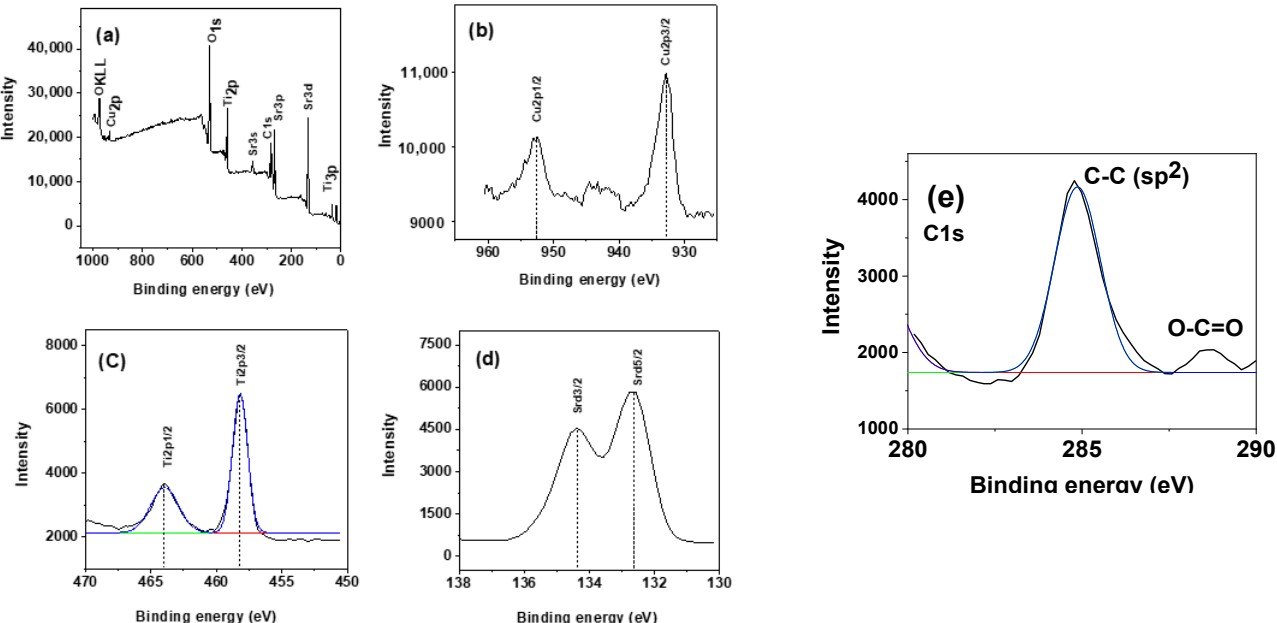

**Figure 3.** XPS survey spectra of 5% CuO/STO/MWCNTs (**a**), high-resolution spectra of Cu 2p (**b**), Ti 2p (**c**), Sr d spectra (**d**), and C 1s (**e**).

TEM images of STO, 5% CuO/STO, and 5% CuO/STO/MWCNTs were taken to understand their morphology, size, and distribution of constituents. As shown in Figure 4a,b, the STO nanoparticles existed in the size of approximately 15 nm. The morphology of 5% CuO/STO/MWCNTs is shown in Figure 4c, where the MWCNTs remained in nanosized tubular shape, and the size of tubular was approximately 25 nm wide. From Figure 4d, it can be seen that some nanoparticles were firmly attached to the wall of the carbon nanotube. These nanoparticles can be considered as CuO/STO in the nanocomposites.

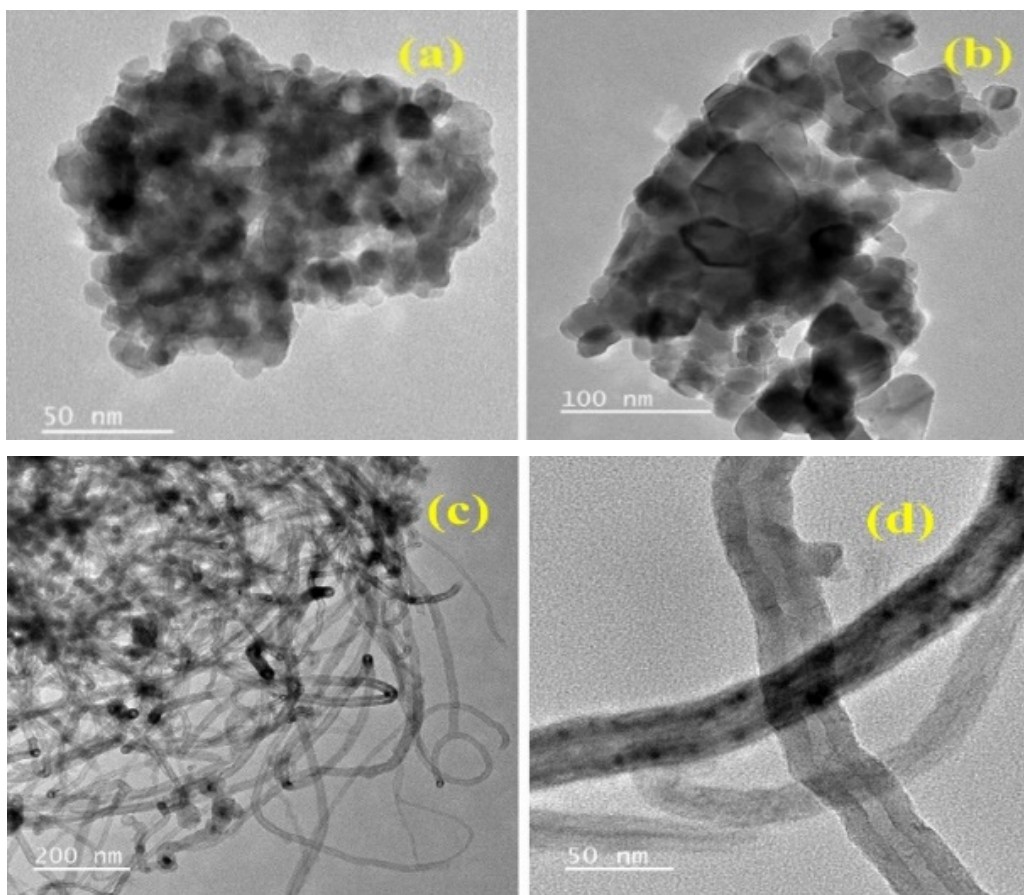

**Figure 4.** TEM of STO (**a**), 5% CuO/STO (**b**), and 5% CuO/STO/MWCNTs (**c**,**d**).

Figures 5 and 6 show the images of DF-STEM-EDX element mapping of the 3% CuO/STO/MWCNTs and 5% CuO/STO/MWCNTs, respectively. The respective elements detected in EDX-mapping are illustrated as the colored spots in the mapping images. As shown in Figures 5 and 6, it is obvious that Cu, Sr, Ti, and O were homogeneously dispersed on the MWCNTs. It is noticed that, because the DF-STEM-EDX mapping was recorded on a JEM-2100 electron microscope, using a copper sample holder led to the colored spots in the mapping images of the Cu element being denser than that of other elements. Combined with the results of TEM and DF-STEM-EDX mapping images, it can be seen that the CuO/STO was loaded on the MWCNTs.

Figure 7 shows the DRS spectra of the STO particles, STO/MWCNTs, and CuO/STO/MWCNTs nanocomposites. Because of its wide band gap, STO exhibited a strong absorption only in the UV light region. In sharp contrast, it can be seen that the nanocomposites display continuously improved visible light absorbance (red shift). The CuO/STO/MWCNTs nanocomposites exhibited further improved visible light absorption, probably due to the strong light absorption of CuO and MWCNTs in the visible light region. The UV–Vis spectra clearly indicated the effect of the presence of CuO and MWCNTs on the visible absorption of the CuO/STO/MWCNTs nanocomposites. Similar red shift has also been confirmed by previous investigators [12,39].

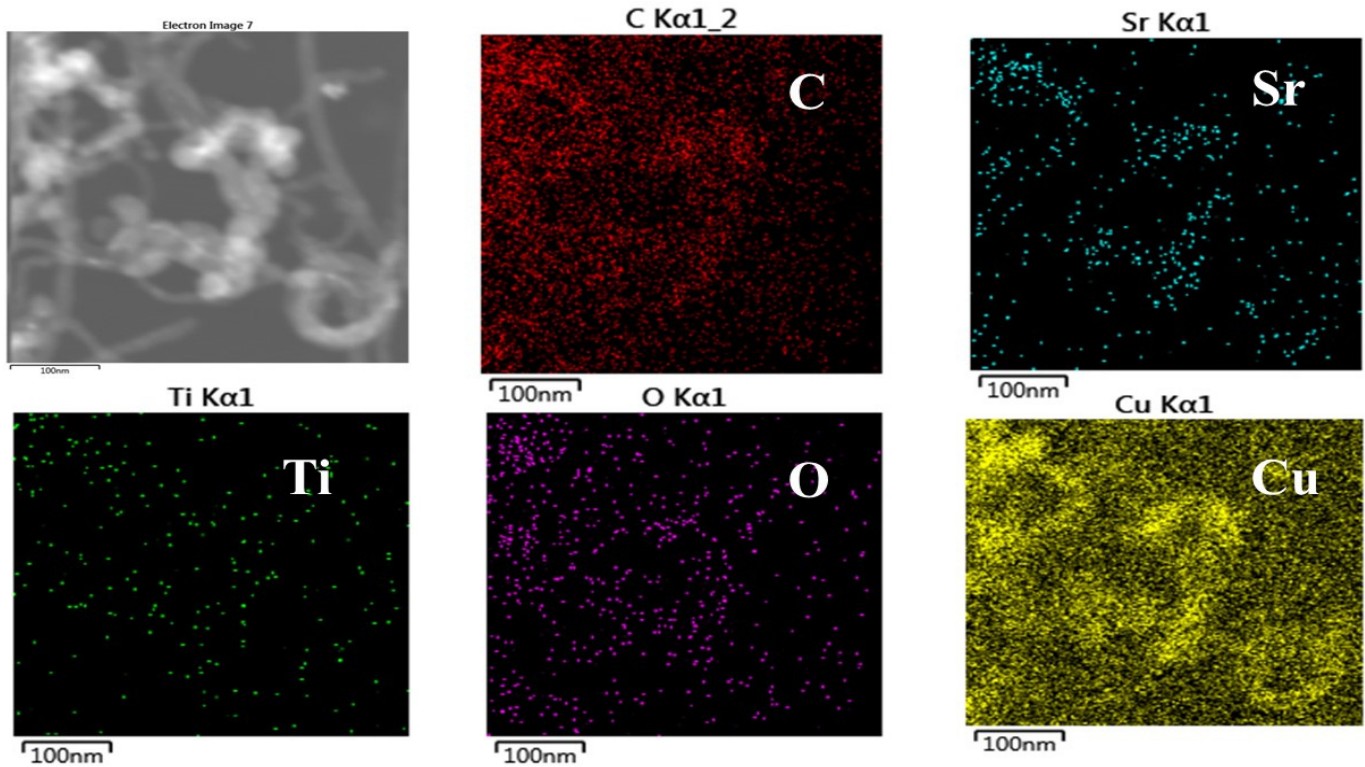

**Figure 5.** DF-STEM micrographs and EDX element mapping images of 3% CuO/STO/MWCNTs.

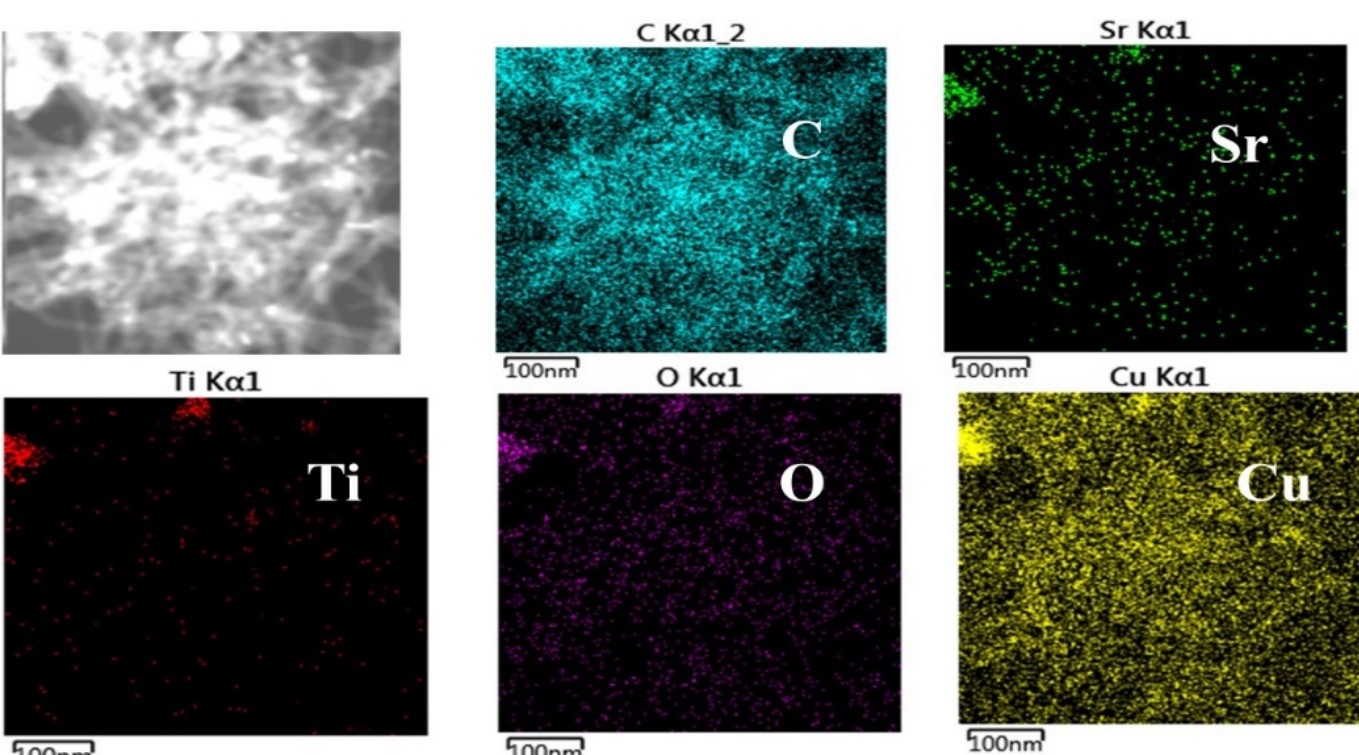

**Figure 6.** DF-STEM micrographs and EDX element mapping images of 5% CuO/STO/MWCNTs.

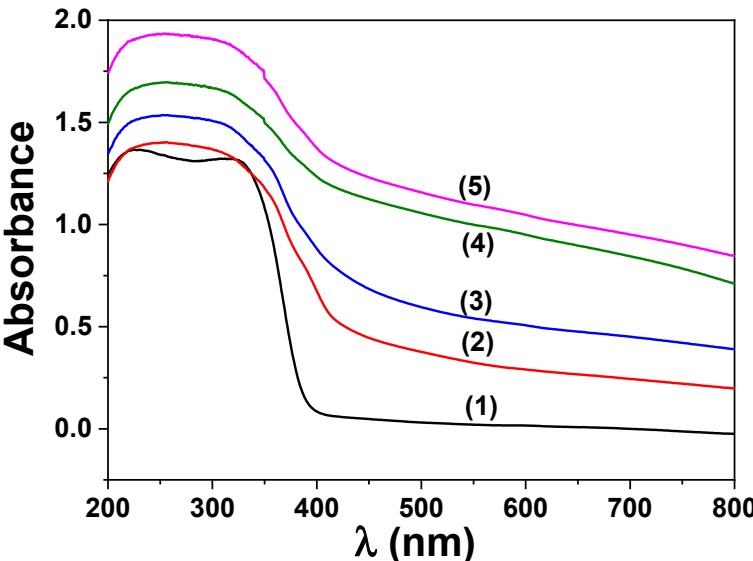

**Figure 7.** DRS spectra of STO (**1**), STO/MWCNTs (**2**), 1% CuO/STO/MWCNTs (**3**), 3% CuO/STO/MWCNTs (**4**), and 5% CuO/STO/MWCNTs (**5**).

The band gap energy has a significant parameter effect on the photocatalytic activity of catalysts. Thus, the band gap energy ($E_g$) of the as-prepared samples was calculated by their UV−Vis optical absorption spectrum and presented in the inset of Figure 8 [40]. As shown in Figure 8, the band gap values of the pure STO, STO/MWCNTs, and x% CuO/STO/MWCNTs (x = 1, 3, 5) samples were found to be 3.23, 2.86, 2.74, 2.38, and 2.25 eV, respectively. Generally, the band gap energy value of the nanocomposites decreased with increasing CuO content.

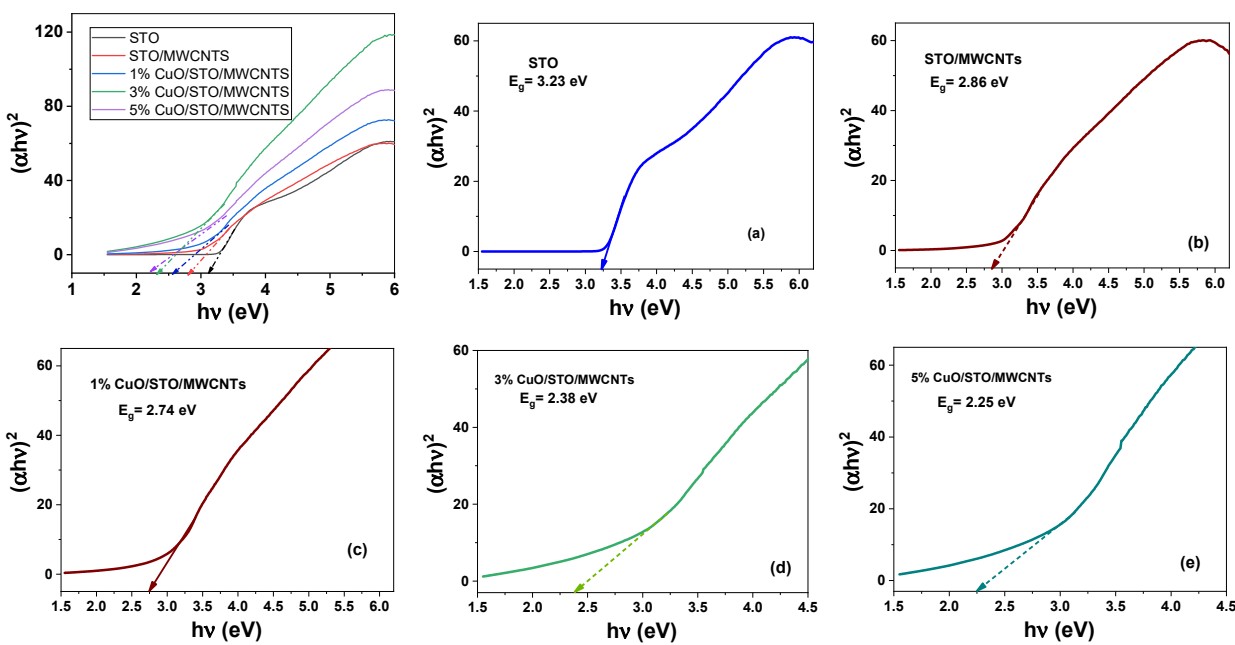

**Figure 8.** The band gap energy of STO (**a**), STO/MWCNTs (**b**), 1% CuO/STO/MWCNTs (**c**), 3% CuO/STO/MWCNTs (**d**), 5% CuO/STO/MWCNTs (**e**).

### 2.2. Photocatalytic Activity of the x% CuO/STO/MWCNTs

To investigate the adsorption performance of the materials, the beaker was placed in the dark with constant magnetic stirring. As shown in Figure 9a, after 30 min, the

adsorption/desorption equilibrium was reached. The adsorption percentages (H%) were about 10% under dark adsorption (Figure 9b).

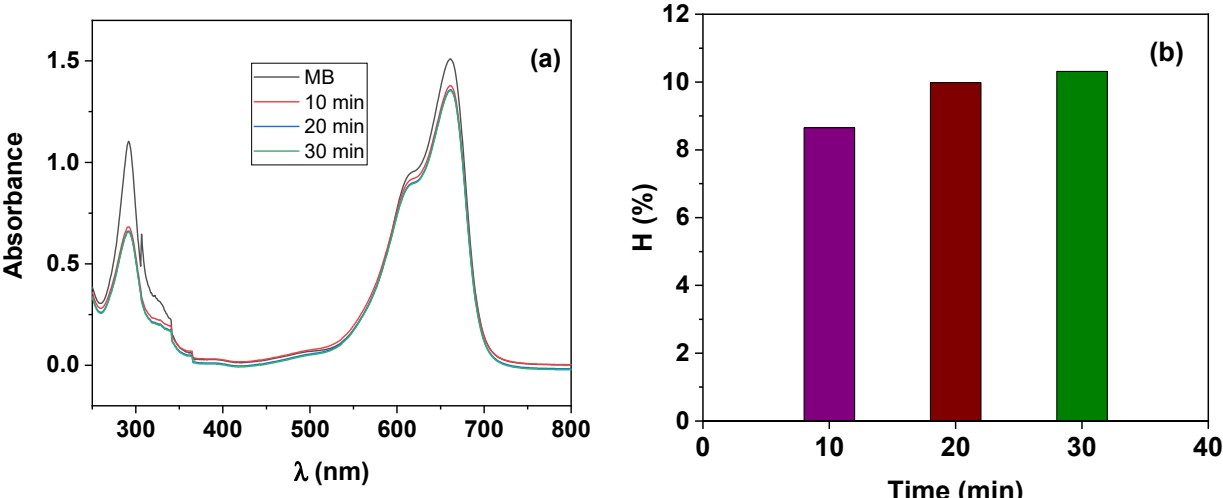

**Figure 9.** UV–vis absorbance spectra of MB under dark adsorption (**a**) and adsorption percentages of 5% CuO/STO/MWCNTs (**b**).

The photodegradation efficiencies of CuO/STO/MWCNTs for the degradation of MB are shown in Figures 10 and 11 and Table 1. It can be seen that all the CuO/STO/MWCNTs exhibited higher photodegradation efficiency than pure STO. This is explained by the fact that, in the case of the CuO/STO-loaded MWCNTs nanocomposites, the MWCNTs and CuO absorb visible light, and the electrons are excited, thereupon generating electron and hole pairs. The photogenerated electrons on the MWCNTs transfer to the surface of the CuO nanoparticles through STO or directly to carry out the photodegradation reaction. Moreover, as the electrons were captured by CuO, the recombination of the electron–hole pairs in the MWCNTs or STO was depressed. Thus, the photocatalytic efficiency of the CuO/STO/MWCNTs was higher than that of STO. Additionally, the enhancement of the photocatalytic activity of the CuO/STO/MWCNTs was presumably explained by the improvement in the absorption of MWCNTs dye, due to their large surface area and special aperture structure.

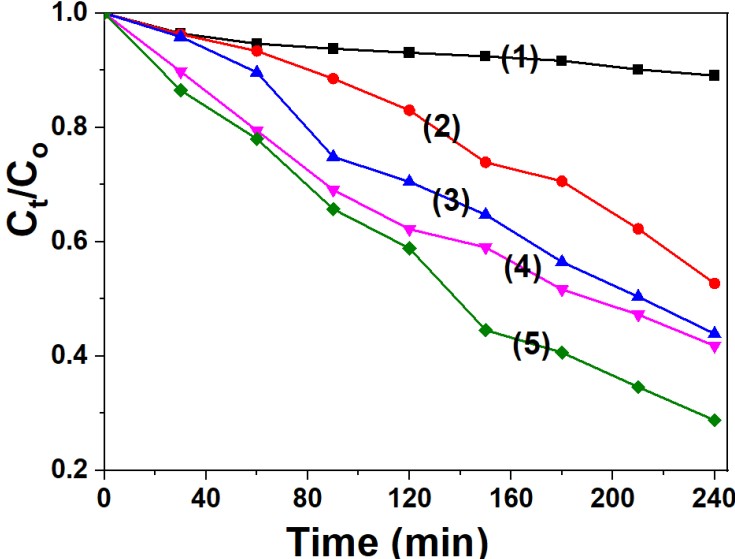

**Figure 10.** Effect of different catalysts on photocatalytic degradation of MB solution at different reaction times STO (**1**), STO/MWCNTs (**2**), 1% CuO/STO/MWCNTs (**3**), 3% CuO/STO/MWCNTs (**4**), 5% CuO/STO/MWCNTs (**5**).

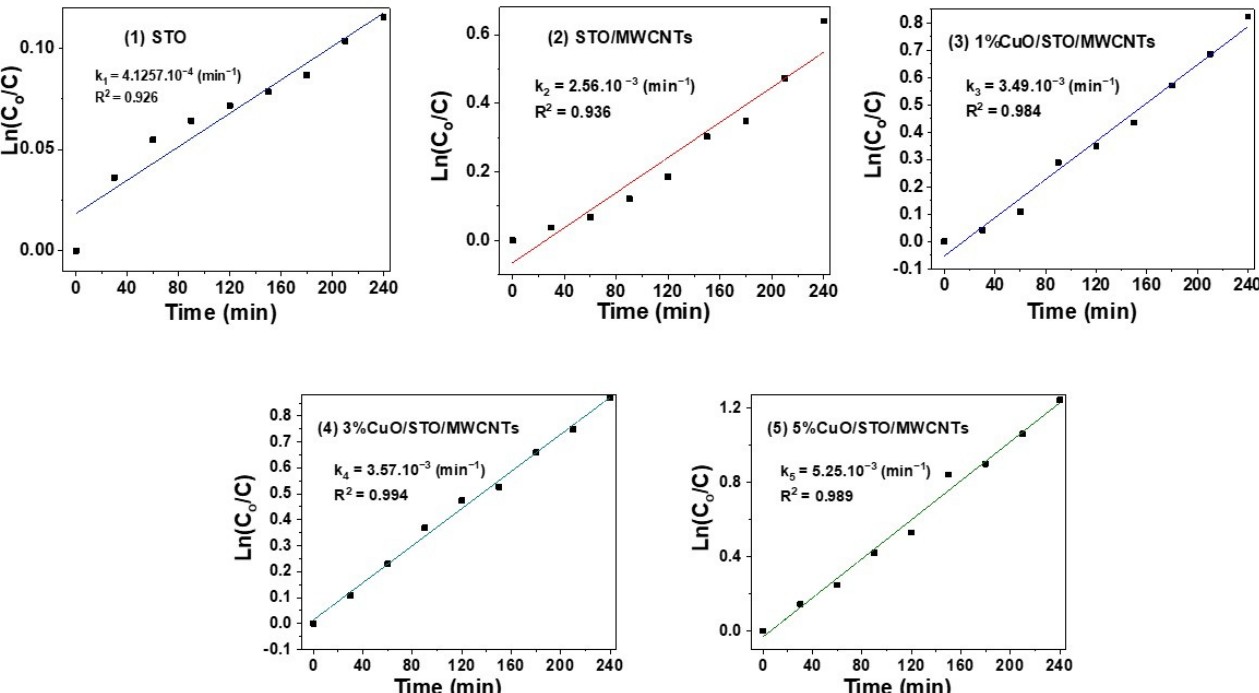

**Figure 11.** The plots of $\ln(C_o/C_t)$ in the presence of (**1**) STO, (**2**) STO/MWCNTs, (**3**) 1% CuO/STO/MWCNTs, (**4**) 3% CuO/STO/MWCNTs, (**5**) 5% CuO/STO/MWCNTs.

**Table 1.** The photodegradation efficiency (H %) and pseudo-first-order rate constant (k).

| Catalysts | H (%) | k (min$^{-1}$) | R$^2$ |
|---|---|---|---|
| STO | 10.91 | $0.0412 \times 10^{-2}$ | 0.926 |
| STO/MWCNTs | 47.26 | $0.256 \times 10^{-2}$ | 0.936 |
| 1% CuO/STO/MWCNTs | 56.08 | $0.349 \times 10^{-2}$ | 0.984 |
| 3% CuO/STO/MWCNTs | 58.19 | $0.357 \times 10^{-2}$ | 0.995 |
| 5% CuO/STO/MWCNTs | 71.22 | $0.525 \times 10^{-2}$ | 0.989 |

As shown in Table 1, the photodegradation efficiency of x% CuO/STO/MWCNTs increased from 56.08 to 71.22%, with a growth of the CuO content from 1 wt.% to 5 wt.%. Based on the results of DRS spectra, it can be explained that the band gap energy value of the CuO/STO/MWCNTs nanocomposites decreased when increasing the CuO content from 1 wt.% to 5 wt.% creating the increasing visible light absorption of the nanocomposites. Therefore, the photocatalytic efficiency of the CuO/STO/MWCNTs for the degradation of MB dye under visible light irradiation was enhanced. On the other hand, CuO could act as activated sites of photocatalytic reactions, when CuO content in the nanocomposites increased, encouraging an increase of the photogenerated carriers for the photocatalytic reactions. As a result, the photocatalytic efficiency of the x% CuO/STO/MWCNTs for the degradation of MB was increased with the growth of the CuO content from 1 wt.% to 5 wt.%. It should be noted that, in the CuO/STO/MWCNTs, nanocomposites was shown to contain a small quantity of $SrCO_3$ impurity (as shown in Figure 1). However, to the best of our knowledge, the $SrCO_3$ do not have significant photocatalytic activity.

Figure 11 depicts the plots with different pseudo first order rates, which were obtained by fitting the following equation:

$$\ln\left(\frac{C_o}{C_t}\right) = kt$$

where $C_o$ is the MB concentration at adsorption equilibrium state, and $C_t$ is the MB concentration at time t after irradiation. k is the pseudo-first-order rate constant.

With the higher coefficient of determination ($R^2 > 0.9$), the proposed model demonstrated high compatibility. Table 1 shows the estimated parameters, including the pseudo-first-order rate constant k and $R^2$ values. The first order rate constants of 5% CuO/STO/MWCNTs and pure STO were $0.525 \times 10^{-2}$ min$^{-1}$ and $0.0412 \times 10^{-2}$ min$^{-1}$. It can be seen that the first order rate constant of 5% CuO/STO/MWCNTs was 12.7 times faster than that of pure STO. Besides, the high $R^2$ values of the samples also confirmed the adherence of the photocatalytic reaction of MB degradation to the first-order kinetics.

The photodegradation efficiency of the 5% CuO/STO/MWCNTs was compared with other competitors and similar materials. The comparative results are shown in Table 2. These results verified that the 5% CuO/STO/MWCNTs can be used for photocatalytic applications, especially for dye degradation under visible light.

To study the recyclability of the photocatalysts, the used 5% CuO/STO/MWCNTs nanocomposites were collected, washed, and dried at 100 °C for 4 h in preparation for reuse. Afterward, the 5% CuO/STO/MWCNTs nanocomposites were recycled three times under similar reaction conditions. As demonstrated by Figure 12, the photocatalytic efficiency for the degradation of MB of the 5% CuO/STO/MWCNTs nanocomposites remain unaltered when the photocatalysts is recycled, and about 99.4% of the initial photocatalytic efficiency was maintained (as shown in Table 1 and Figure 12d).

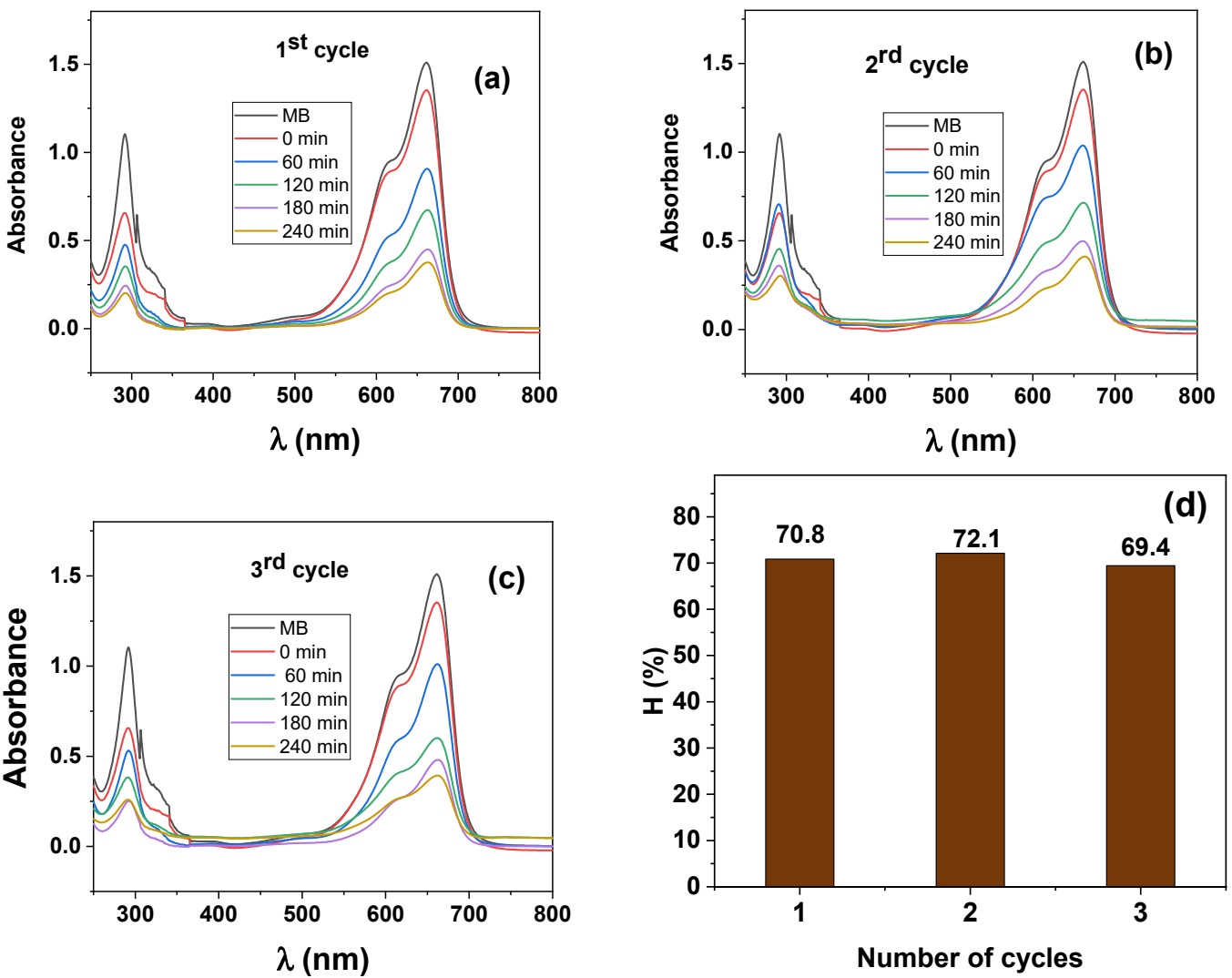

**Figure 12.** The recyclability of the 5% CuO/STO/MWCNTs for degradation of MB in 1st cycle (**a**), 2rd cycle (**b**), 3rd cycle (**c**), and the recycling efficiency of the 5% CuO/STO/MWCNTs (**d**).

Finally, in order to clarify the structural changes of the photocatalyst after the photocatalytic reaction, the recovered 5% CuO/STO/MWCNTs photocatalysts (after three cycles were complete) were re-evaluated with the XRD, DRS, and DF-STEM-EDX mapping techniques. The data analysis are given in Figures 13 and 14. It can be observed that the main diffraction peaks (as shown in Figure 13a), the visible light absorption probably (as shown in Figure 13b), and the elemental composition (as shown in Figure 14) of the recycled 5% CuO/STO/MWCNTs nanocomposites were in good accordance with the as-prepared sample. These results verified that the crystallographic structure, optical characteristics, and composition of the 5% CuO/STO/MWCNTs maintained good stability after three catalytic cycles were recycled.

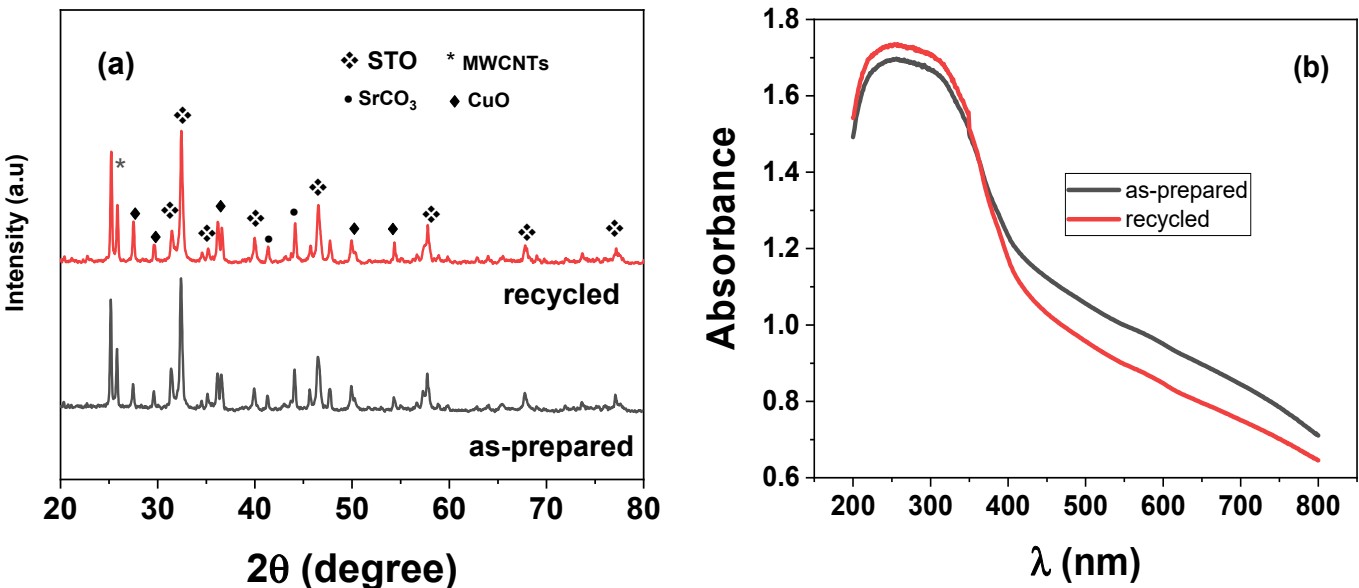

**Figure 13.** XRD patterns (**a**), DRS spectra (**b**) of the as-prepared and recycled 5% CuO/STO/MWCNTs nanocomposites.

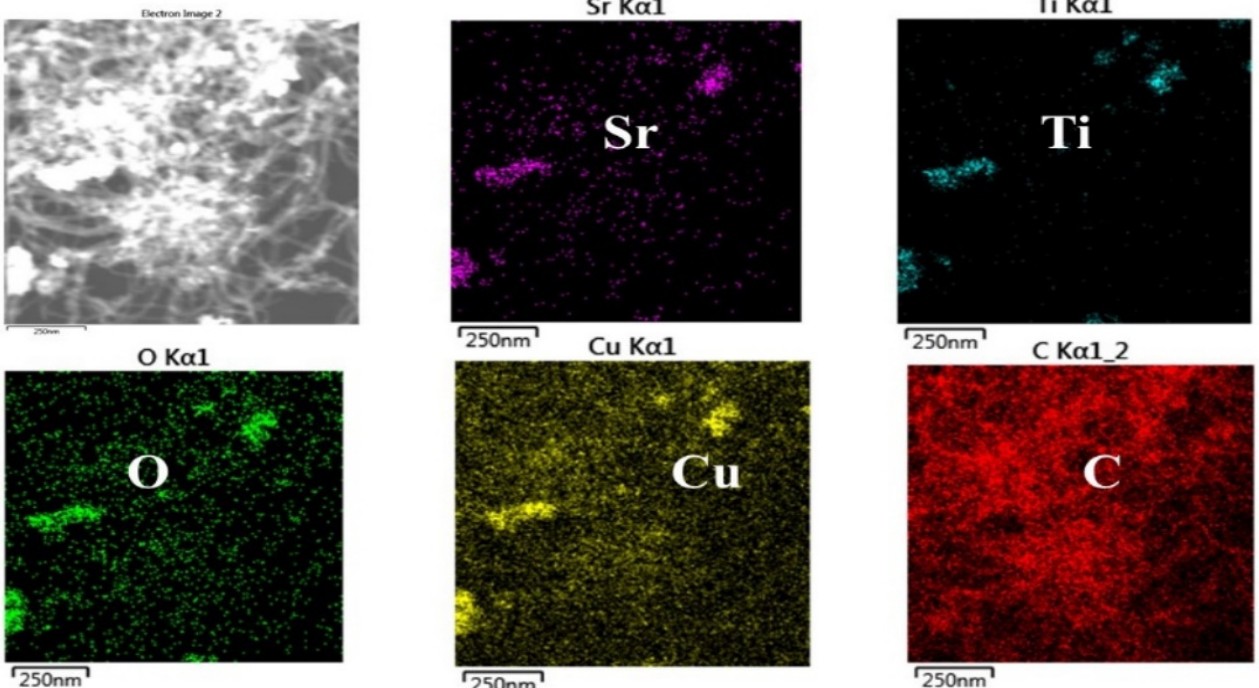

**Figure 14.** DF-STEM-EDX mapping of the recycled 5% CuO/STO/MWCNTs nanocomposites.

To evidence the photodegradation mechanism and the charge recombination, the PL analyses were performed. As shown in Figure 15, the PL intensity of the 5% CuO/STO/MWCNTs nanocomposites was lower than the pure STO. This suggests the existence of a photogenerated electron transfer from STO to CuO, which efficiently reduces the recombination of photogenerated electrons and holes in 5% CuO/STO/MWCNTs nanocomposites. It further confirms that the increasing photocatalytic activity of 5% CuO/STO/MWCNTs nanocomposites is ascribed to a more effective charge separation and the interaction of CuO and STO [41].

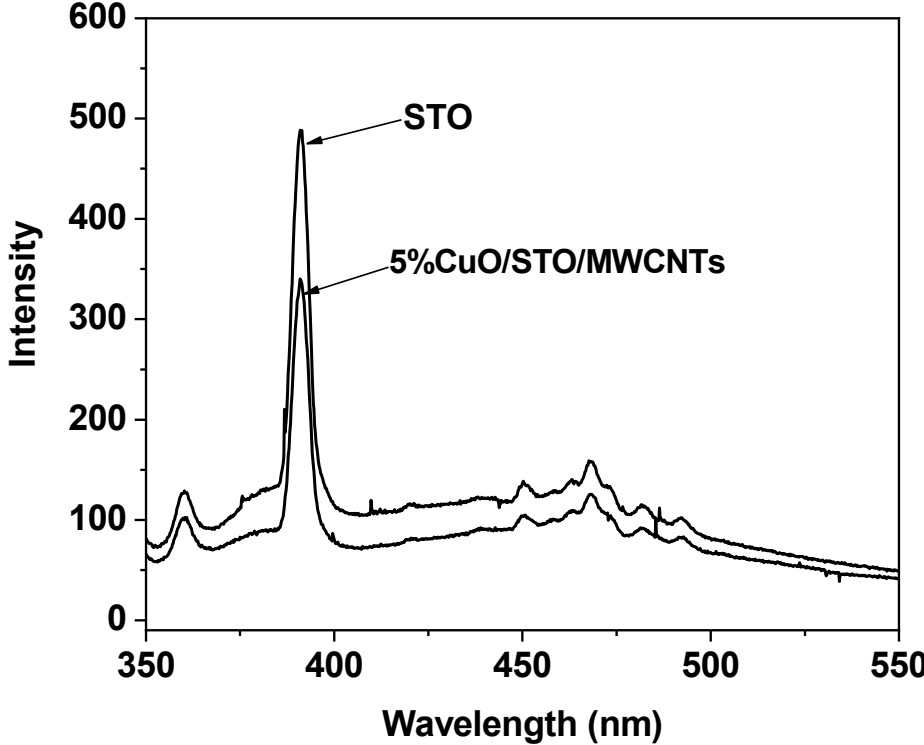

**Figure 15.** PL spectra of STO and 5% CuO/STO/MWCNTs.

Based on the above analysis, for more profound understanding of the effect of CuO loading on the photocatalytic activity of the CuO/STO/MWCNTs photocatalyst for dye degradation under visible light, a possible mechanism has been developed, as shown in Figure 16 [42–44]. Under visible light irradiation, MWCNTs and CuO absorb visible light and the electrons were excited, thereupon generating the photogenerated electrons and holes. The photogenerated electrons on the MWCNTs transfer to the surface of the CuO nanoparticles through STO or directly. Furthermore, the electrons of the VB of STO were induced into MWCNTs, and the positive holes were left in the VB of STO. In other words, the charge of MWCNTs was positive in removing electrons from the VB of the STO to form holes in the VB of STO. The holes on the VB of STO transferred to the VB of CuO or reacted with the water molecules and/or MB dye to generate the hydroxyl radicals, which were reactive enough to result in the photodegradation of MB [45,46].

In the photocatalytic system, under visible light irradiation, MWCNTs acted as photosensitizers that got excited, and the CuO acted as co-catalyst and charge transferring sites and/or active sites in the photocatalytic process. Since the electrons were captured by CuO, the recombination of electrons and holes in the MWCNTs and STO was depressed, which makes for improving the separation of photogenerated electron–hole pairs and increasing the photocatalytic activity of STO. It was noticed that, in the case of these nanocomposites, not only CuO can generate active centers, the electrons and holes on the surface of MWCNTs and STO could react with the $O_2$ and $H_2O$ of the solution to form OH· radicals, and others, which are powerful oxidatives for carrying out the photodegradation of MB dye.

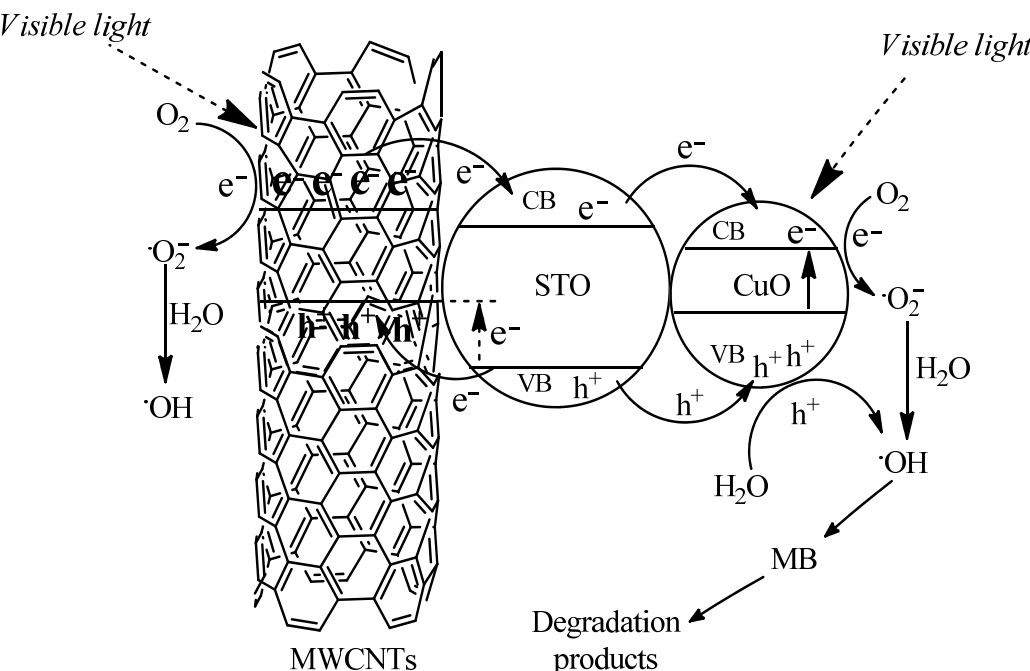

**Figure 16.** The mechanism for photodegradation of MB by CuO/STO/MWCNTs nanocomposites under visible light irradiation.

**Table 2.** Comparison of photocatalytic degradation efficiency of 5% CuO/STO/MWCNTs nanocomposites with other competitors and similar materials.

| Catalysts | Organic Pollutants | Conditions | H (%) | Ref. |
|---|---|---|---|---|
| CuO/rGO (graphene) | MB | MB ($2.23 \times 10^{-5}$ M), catalyst (1 g L$^{-1}$), 100 W Xe lamp, 90 min. | 77.8% | [29] |
| STO/rGO (graphene) | CIPRO (ciprofloxacin) | CIPRO (5 mg L$^{-1}$), catalyst (1 g L$^{-1}$), 40 W compact fluorescent lamp, 7 h. | 74.1% | [47] |
| | IBP (Ibprofen) | IBP (5 mg L$^{-1}$), catalyst (1 g L$^{-1}$), 40 W compact fluorescent lamp, 7 h. | 68.4% | |
| | RhB | RhB (5 mg L$^{-1}$), catalyst (1 g L$^{-1}$), 40 W compact fluorescent lamp, 7 h. | 100% | |
| CNTs/Ag/TiO$_2$ | MB | MB (10 mg L$^{-1}$), catalyst (1 g L$^{-1}$), halogen lamp, 3 h. | 80.8% | [24] |
| 5% CuO/STO/ MWCNTs | MB | MB (10 mg L$^{-1}$), catalyst (1 g L$^{-1}$), 500 W halogen lamp, 3 h. | 71.2% | Present work |

## 3. Materials and Methods

### 3.1. Materials

All chemicals were analytical grade and employed without being further purified. Strontium nitrate (Sr(NO$_3$)$_2$), tetra-butyl titanate TBOT (Sigma-Aldrich, St. Louis, MO, USA), Cu(NO$_3$)$_2$ (Xilong Chemical Co., Ltd. Guangzhou, China). Functional multi-walled carbon nanotubes modified with carboxyl groups (MWCNTs-COOHs) were purchased from Chengdu Organic Chemical Co, Ltd. (Chengdu, China).

### 3.2. Preparation of STO Nanoparticles

The STO nanoparticles were initially fabricated by sol–gel method, and the experimental procedures were discussed as follows. First, 2.12 g of Sr(NO$_3$)$_2$ and 2.10 g of citric acid were added into 40 mL of deionized water. The pH solution was controlled to about

8 with aqueous ammonia (denoted as solution A). On the other hand, 3.40 g of TBOT was dissolved into 20 mL of ethanol (denoted as solution B). Then, drop of solution B was added into solution A under vigorous stirring. The obtained solution was stirred at 70 °C. After, the obtained solution was evaporated and dried at 80 °C for 12 h to form a xerogel. Eventually, the obtained sample was calcined at 700 °C for 4 h.

### 3.3. Preparation of CuO/STO Nanoparticles

The STO nanoparticles, which were loaded with various amounts of CuO (1, 3, 5 wt.%), were prepared by impregnation method, followed by heat treatment. A typical experimental procedure of 1 wt.% CuO/STO was reported in the following process: 200 mg of the as-prepared STO and 2.5 mL of $Cu(NO_3)_2$ 0.01 mol·$L^{-1}$ solution were initially injected into 10 mL distilled water. Next, the solution was stirred for 4h at room temperature. Sample was then dried at 80 °C for 12 h and followed by calcination for 4 h at 450 °C. The other CuO/STO samples were synthesized using similar procedures. The synthesized samples were named x% CuO/STO (which, x% was CuO weight percent).

### 3.4. Preparation of STO/MWCNTs Nanoparticles

The STO nanoparticle-loaded MWCNTs (10 wt.%) were prepared by mixing in aqueous medium methods: 30 mL distilled water was added into 200 mg of as-prepared STO and 20 mg of functional MWCNTs. The resulting mixture was slowly stirred for 1 h. In the final stage, the sample was further dried at 80 °C for up to 12 h. The obtained products were denoted as STO/MWCNTs.

### 3.5. Preparation of CuO/STO/MWCNTs Nanocomposites

The CuO/STO-loaded MWCNTs (CuO/STO/MWCNTs) were prepared mixing in aqueous medium methods. A typical experimental procedure of 1% CuO/STO-loaded MWCNTs was narrated as follows: 20 mg of functional MWCNTs (10 wt.%) and 200 mg of as-prepared 1% CuO/STO were added into 30 mL distilled water. The resulting mixture was slowly stirred for 1 h. Finally, the resulting mixture was dried at 80 °C for up to 12 h. The other CuO/STO-loaded MWCNTs samples were synthesized using similar procedures. The samples were denoted as x% CuO/STO/MWCNTs (which x% was CuO weight percent).

### 3.6. Characterization

The physicochemical properties of the as-prepared samples were determined based on the use of the following instruments: The structure and crystalline phase of the as-prepared samples were characterized by X-ray powder diffractometer (D8 Advance, Brucker, Madison, WI, USA). The microstructure, morphology, elemental composition of the samples (TEM, DF-STEM-EDX mapping) were analyzed by JEM-2100 electron microscope (JEOL Ltd., Tokyo, Japan), energy dispersive X-ray spectroscopy (EDX, Hitachi S-4800, Tokyo, Japan), solid diffusion reflectance UV–Vis spectra (DRS, Hitachi U-4100, Tokyo, Japan). The X-ray photoelectron spectra (XPS) of samples were recorded on a PHI 5000 Versaprobe spectrometer (ULVAC-PHI, Inc., Kanagawa, Japan). The photoluminescence spectra (PL) of samples were taken with a RF-5301PC fluorescence spectrometer (Shimadzu, Tokyo, Japan).

### 3.7. Photocatalytic Experiments

The photocatalytic activity of the photocatalyst was investigated by the degradation of methylene blue (MB) dye aqueous solution in the visible light range. Especially, 100 mg of photocatalyst was dispersed into 100 mL aqueous solution of MB (10 mg $L^{-1}$) and followed by stirring without any light source for 30 min to obtain adsorption/desorption equilibrium. Afterward, the suspension was irradiated by a 500 W halogen lamp (Philips, Amsterdam Netherlands) with a cooling water jacket. For a typical photodegradation experiment, after every 30 min irradiation, 5 mL of suspension was withdrawn and centrifuged to remove the photocatalyst particles. Finally, the transparent solution was analyzed by a UV–Vis

spectrophotometer (Shimadzu UV-1700, Tokyo, Japan). To investigate the recyclability of photocatalyst, the used 5% CuO/STO/MWCNTs nanocomposites were collected, washed with distilled water, and dried at 100 °C for 4 h in preparation for its reuse. The photocatalytic reactions were recycled three times. The photodegradation efficiency was calculated using the following equation:

$$H = \frac{C_o - C_t}{C_o} \times 100\%$$

where $C_o$ and $C_t$ are the concentration of MB (mg.L$^{-1}$) at time 0 and t and at $\lambda$ = 663 nm.

### 4. Conclusions

In the present study, the CuO/STO/MWCNTs nanocomposites were synthesized successfully by a sequential three-step process of sol–gel, impregnation, and mixing via aqueous methods. The photocatalytic activity of the CuO/STO/MWCNTs nanocomposites were studied and discussed for the degradation of MB under visible light irradiation. The as-prepared CuO/STO/MWCNTs photocatalyst demonstrated higher photocatalytic efficiency than the pure STO. The photodegradation efficiency of x% CuO/STO/MWCNTs increased from 56.08 to 71.22% when increasing the CuO content from 1 wt.% to 5 wt.%. The main role of the CuO in the CuO/STO/MWCNTs nanocomposites was studied. It can be seen that the nanocomposites continuously displayed enhanced light absorbance over the whole wavelength range, with the increasing loaded CuO content and the photodegradation efficiency x% CuO/STO/MWCNTs increasing with increasing CuO content. The increased photocatalytic activity of x% CuO/STO/MWCNTs, due to the value of band gap energy of the x% CuO/STO/MWCNTs nanocomposites, decreased with increasing CuO content, leading to increased visible light absorbance. The most potential mechanism of the photocatalytic reaction was recommended. Under visible light irradiation, the MWCNTs acted as photosensitizers, and the loaded CuO acted as co-catalyst and charge transferring sites and/or active sites in the photocatalytic process. Thanks to the capture of electrons by CuO, the recombination of electron–hole pairs in STO was depressed. The process led to the enhancement of the separation of photogenerated electron–hole pairs, suppressing recombination between electrons and holes, and improving charge migration and photocatalytic activity of STO. Finally, the 5% CuO/STO/MWCNTs showed excellent stability and recyclability to maintain the initial activity after three consecutive catalytic cycles, and the crystallographic structure, optical characteristics, and composition of the 5% CuO/STO/MWCNTs remained unaltered when the photocatalysts was recycled.

**Author Contributions:** Conceptualization, X.T.M. and V.K.P.; methodology, T.H.T.P.; software, T.T.L.N.; formal analysis, H.D.C. and T.H.T.P.; data curation, T.T.L.N. and X.T.M.; writing—original draft preparation, D.N.B. and V.K.P.; writing—review and editing, D.N.B. and T.K.N.T. All authors have read and agreed to the published version of the manuscript.

**Funding:** This work was supported by the MOET of Vietnam project No. B2020-TNA-12.

**Institutional Review Board Statement:** Not applicable.

**Informed Consent Statement:** Not applicable.

**Data Availability Statement:** All the data are available within the manuscript.

**Conflicts of Interest:** The authors declare no conflict of interest.

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
