# Peer review of "Effect of CuO Loading on the Photocatalytic Activity of SrTiO3/MWCNTs Nanocomposites for Dye Degradation under Visible Light"

_inorganics, doi:10.3390/inorganics10110211_

Round 1

Reviewer 1 Report

This work presents the synthesis of photocatalytic nanocomposites made of CuO/STO/MWCNTs. Researchers present complete characterization of the materials and their properties, then test them for the degradation of dyes under the visible light spectrum. It was interesting research - I would recommend comparing the materials MB degradation with those of other perovskite nanocomposites. 

Authors should consider the following: 

1. How cost effective in terms of time and pricing is the CuO/STO/MWCNTs photocatalysts vs other similar systems? How easily can it be integrated into MB degradation systems?

2. EDX data should be plotted on origin/graphing software and the data graphs from the software itself should be placed in the SI

3. Why use TEM over SEM here? I would add a sentence to explain. 

4. Fig 7-9 should be redone to be more legible. 

5. Revise Fig 11 - with the added red bars it is not very legible; it can be made more clear to the reader

6. I would plot or add a table comparing the 5% CuO/STO/MWCNTs efficiency at MB degradation with other competitors and similar materials

7. What method was used to prepare the bought MWCNTs-COOH? Does it say on the label? And why did you choose them?

Author Response

Dear Editor and reviewers,

We would like to express our gratitude for the Editor and Reviewer’s efforts to improve the quality of our manuscript. We have tried our best to respond to all issues indicated in the review report sufficiently. In the revised version, we have highlighted the changes to our manuscript using the red color. The answers to the questions you raised are detailed here.

Reviewer 2 Report

The authors reported the effect of CuO loading on the photocatalytic activity of SrTiO3/MWCNTs nanocomposites for Dye degradation, this is an interesting work, however, some revisions are needed before it be accepted.

1.       In lines 99-105, the authors introduction the photo-degradation activity of STO-graphene composites, what’s the relationship between STO-graphene and MWCNTs.

2.       In lines 177-178, the authors declare that the size of STO is about 15 nm, so, what’s the sizes of CuO nanoparticles and MWCNT? Besides, please provide the TEM images of CuO and 5%CuO/STO composites.

3.       According to the author’s opinion, 5%CuO/STO/MWCNTs exhibits the optimum photodegradation performance, how about 7%CuO/STO/MWCNTs,  9%CuO/STO/MWCNTs? Do you think the higher the CuO content in the composites, the better of the photocatalytic activity?

4.       To illustration photodegradation mechanism, BET, I-t, PL, TRPL etc need to be added.

5.       The reference format needs to be consistent.

6.       It’s suggested the authors correct the grammatical mistakes in the manuscript. 

Author Response

(The authors gave the same response as above.)

Reviewer 3 Report

The current version of the manuscript does not look worthy and cannot be recommended for publication in this form. The manuscript has several lacks in the explanation and experiment. The novelty of the paper is not apparent. In addition, the quality of the present form of this paper is not commensurate with the high standards of the journal Inorganics. The following aspects need improvement, and the comments are expected to be discussed in the text before publication.

1. The authors should carefully check the typo and space errors throughout the manuscript.

2. The abstract is not clearly written. It should be rewritten to reflect the scientific and technological importance of the work without introducing the characterization techniques here in the abstract.

3. There are lots of articles already published using CuO/MWCNTs material and showed very good results compared to the present work. Then why SrTiO3 is important although the activity is low? The authors must be addressed the novelty of the work.

4. There are so many unidentified peaks in the XRD spectra. It should be explained. In addition, the authors used and * for indicating the same compound (SrCO3) peaks. How is it possible? The compound name (SrCO3) should be checked.  

5. Figure 3 caption is wrong. It should be 5% CuO/STO/MWCNTs. Also, the carbon spectrum must be explained. Figure 3c needs to show the deconvolution peaks.

6. Data analysis in Figure 5 needs serious discussion and justification. It looks unusual. See the latest article by the editors of “Optical Materials” journal:
Brik, M. G., Srivastava, A. M., & Popov, A. I. (2022). A few common misconceptions in the
interpretation of experimental spectroscopic data. Optical Materials, 127, 112276.

7. In section 3.3, lines 325-326, they wrote “The other CuO/STO samples were prepared using similar procedures. The obtained samples were named x% CuO/STO (which x% was CuO weight percent). But there is no other sample’s data in the manuscript. Also, only XRD data is explained. So, why it is important?

8. They explained STO/MWCNTs. But there is no preparation process in the manuscript. It must be discussed.

9. The title and section 3.4 (Preparation of CuO/STO/MWCNTs nanocomposites) are totally mismatched. It must be clarified CuO loaded STO/MWCNTs or CuO/STO loaded MWCNTs.

10. It is highly recommended to investigate scavenger experiments for specifying which radical/radicals are responsible for the degradation of the pollutants.

11. The authors must be compared their photocatalytic activity with the published work. Without comparing how do the readers know your product is better?

12. The authors should carry out LC-MS so that the possible degradation pathway and possible degradation products are specified.

13. The pH is an important factor during the degradation process that must be considered. It is highly recommended to investigate the point of zero charges (PZC) for their best composite and explain the results with pH experiments.

14. Adsorption tests (without light) are missing. Please include and discuss the adsorption performance of the materials. Since carbon materials, in general, are good adsorbents.

15. The stability and reusability of the compound are not present, which is very important for the practical application of a catalyst. Therefore, the authors should explain the reusability performance of the catalyst. Also, the authors should clarify the structural changes of the catalyst after the photocatalytic reaction.                        

Author Response

(The authors gave the same response as above.)

Round 2

Reviewer 3 Report

The authors addressed some issues but many of the comments are still not explained properly. Therefore, the following corrections are needed before accepting. 

1. There are so many unidentified peaks in the XRD spectra. It should be explained.

2. Revise lines 174 to 182. The explanation of the Figure should be written in a sequence that means Figure a, b, c, d, and e.

3. What do you mean by blue spots inside Figure 3c?

4. In Figure 3e, the authors should explain all the deconvoluted peaks not only one peak.

5.  Adsorption percentages should be addressed.

6. Checked the title and this line (CuO/STO loaded MWCNTs). It is mismatched. Revised the title.

7. There is no relevant data regarding stability and reusability experiments. Without data, only an explanation is not enough. I strongly recommended adding this valuable data.

8. The authors must be clarified the structural changes of the catalyst after the photocatalytic reaction.

Author Response

Dear Editor and reviewers,

We would like to express our gratitude for the Editor and Reviewer’s efforts to improve the quality of our manuscript. We have tried our best to respond to all issues indicated in the review report sufficiently. In the revised version, we have highlighted the changes to our manuscript using the blue color. The answers to the questions you raised are detailed here. In addition, some sentences in the manuscript have been rewritten to reduce the similarity index.

Round 3

Reviewer 3 Report

The manuscript has significantly improved in its quality. Hence, it is recommended to accept for publication in this journal.